# A Multidisciplinary Diagnostic Approach Reveals a Higher Prevalence of Indolent Systemic Mastocytosis: 15-Years’ Experience of the GISM Network

**DOI:** 10.3390/cancers13246380

**Published:** 2021-12-20

**Authors:** Roberta Zanotti, Massimiliano Bonifacio, Cecilia Isolan, Ilaria Tanasi, Lara Crosera, Francesco Olivieri, Giovanni Orsolini, Donatella Schena, Patrizia Bonadonna

**Affiliations:** 1Hematology Unit, Department of Medicine, Azienda Ospedaliera Universitaria Integrata di Verona, 37134 Verona, Italy; roberta.zanotti@univr.it (R.Z.); ceci.odilla.iso@gmail.com (C.I.); ilaria.tanasi@univr.it (I.T.); lara.crosera@gmail.com (L.C.); 2Gruppo Interdisciplinare per lo Studio Della Mastocitosi (GISM), Azienda Ospedaliera Universitaria Integrata di Verona, 37134 Verona, Italy; giovanni.orsolini@aovr.veneto.it (G.O.); donatella.schena@aovr.veneto.it (D.S.); patrizia.bonadonna@aovr.veneto.it (P.B.); 3Allergy Unit, Department of Medicine, Azienda Ospedaliera Universitaria Integrata di Verona, 37134 Verona, Italy; olivieri.associazionerima@gmail.com; 4Rheumatology Unit, Department of Medicine, Azienda Ospedaliera Universitaria Integrata di Verona, 37134 Verona, Italy; 5Dermatology Unit, Department of Medicine, Azienda Ospedaliera Universitaria Integrata di Verona, 37134 Verona, Italy

**Keywords:** systemic mastocytosis, prevalence, incidence, multidisciplinary approach

## Abstract

**Simple Summary:**

The approach of the Interdisciplinary Group for Study of Mastocytosis (GISM) of Verona, based on a regional network of clinical specialists and expert pathologists, with the availability of sensitive molecular and flow-cytometry diagnostic techniques, allowed the discovery of a higher-than-expected number of patients with systemic mastocytosis (SM), demonstrating a prevalence of 17.2/100,000 adult inhabitants. During a 15-year period of activity, we documented a remarkable increase in diagnosis of indolent SM, which constitute over 90% of our SM series, mainly represented by the bone marrow mastocytosis subvariant (54.8%). A timely diagnosis of SM has a great practical importance, since allows discovering and treating underdiagnosed osteoporosis, especially in males, a condition which is frequently complicated with fragility fractures, also in young patients, with disabling consequences. Moreover, diagnosis of mastocytosis in patients with Hymenoptera venom allergy allows continuing lifelong venom immunotherapy in order to prevent further severe, sometimes fatal, allergic reactions.

**Abstract:**

Systemic mastocytosis (SM) and other adult clonal mast cell disorders (CMD) are often underestimated, and their epidemiology data are scarce. We aimed at evaluating the impact of the activity of the Interdisciplinary Group for Study of Mastocytosis (GISM) of Verona on the prevalence and incidence of CMD. We examined the data of 502 adult patients diagnosed with CMD and residing in the Veneto Region, consecutively referred to GISM between 2006 and 2020. SM was diagnosed in 431 cases, while 71 patients had cutaneous mastocytosis or other CMD. Indolent SM represented the most frequent SM variant (91.0%), mainly with the characteristics of bone marrow mastocytosis (54.8%). The prevalence of SM in the adult population of the Veneto region and of the Verona province was 10.2 and 17.2/100,000 inhabitants, respectively. The mean incidence of new SM cases in Verona was 1.09/100,000 inhabitants/year. Hymenoptera venom allergy was the main reason (50%) leading to the CMD diagnosis. Osteoporosis, often complicated by fragility fractures, was present in 35% of cases, even in young patients, especially males. Our data show a higher prevalence and incidence of SM than previously reported, confirming that reference centers with multidisciplinary approach are essential for the recognition and early diagnosis of CMD.

## 1. Introduction

Mastocytosis consists of clonal mast cell (MC) disorder (CMD) characterized by an abnormal proliferation of mast cells infiltrating various tissues, particularly skin and hematopoietic organs. Disease subtypes range from indolent to rare but aggressive forms [1]. Mastocytosis and other CMD may present with a variety of clinical manifestations due to inappropriate release of MC mediators (i.e., pruritus, urticaria, angioedema, flushing, nausea, vomiting, abdominal pain, diarrhea, episodic anaphylactoid attacks, osteopenia, or osteoporosis). Additionally, in the aggressive disease, clinical features are also related to MC tissue infiltration and subsequent organ dysfunction (i.e., hypersplenism, pathological bone fractures, ascites, malabsorption, and cytopenia) [1].

Mastocytosis is divided into two major categories including cutaneous mastocytosis (CM), limited to the skin and more common in pediatric cases, and systemic mastocytosis (SM), in which abnormal MC are present in extracutaneous organ, mainly bone marrow (BM), almost exclusively associated with adulthood [1]. According to the revised 2016 World Health Organization Classification (WHO) diagnostic criteria, mastocytosis is divided into CM, indolent SM (ISM), smoldering SM (SSM), aggressive SM (ASM), SM with an associated hematological neoplasm (SM-AHN), mast cell leukemia (MCL), and mast cell sarcoma (MCS) [2].

Furthermore, the term monoclonal mast cell activation syndrome (MMAS) has been proposed to identify patients with recurrent, severe symptoms from MC-derived mediators in which BM examination fails to demonstrate the minimum diagnostic criteria for SM, but MC clonality is proved [3,4,5,6].

Mastocytosis is a rare disease, but it is also underestimated. In fact, ISM diagnosis, above all in the provisional WHO subvariant without skin lesion (bone marrow mastocytosis–BMM), may easily be missed as symptoms can be subtle or absent, together with the lack of awareness of this disease [7,8].

Mastocytosis can occur at any age [9]. In adults, the median reported age at diagnosis ranges between 50 and 57 years [10,11,12]. No significant difference by sex was documented in one of the largest published studies, but a slightly male or female prevalence was reported in other studies reflecting the different populations referred to each center [10,11,12,13]. Familial cases of mastocytosis were reported in pediatric series and, recently, in a large series of adults with an estimated frequency of 1.5% [14].

Few studies had addressed the issue of incidence and prevalence of mastocytosis in adults, with a reported estimated prevalence of SM of 10–13 patients per 100,000 people [9,11,12]. However, the actual prevalence of mastocytosis may be higher.

The aim of this study was to evaluate the epidemiological data of mastocytosis in the adult (≥15 years) population of the Veneto region (north-east Italy), based on the activity of our Interdisciplinary Group for the Study of Mastocytosis (GISM), established in Verona in January 2006. A deeper evaluation of the epidemiological data was performed in the Verona province, assuming that virtually all patients with suspected mastocytosis living in that area are referred to GISM. We also aimed at determining the “real life” distribution of the different variants as a result of a regional multidisciplinary network. Moreover, we sought whether the diagnoses of mastocytosis had an impact on early detection of bone complications as osteoporosis and fragility fractures [15,16] and on administration of lifelong venom immunotherapy (VIT) in patients with Hymenoptera Venom Allergy (HVA) in order to prevent severe, sometimes fatal, reactions after discontinuation of VIT [5,17,18].

## 2. Materials and Methods

Patients with suspected mastocytosis are referred to GISM from a wide network of Allergological, Dermatological, Rheumatologic, and Hematological centers distributed in the Veneto region, and also from specialists of other Regions of Italy.

In all patients deemed suitable for BM evaluation, the following evaluations were performed: morphologic examination of BM smear, BM biopsy including immunohistochemistry with tryptase, CD117 and CD25 staining, detection of the D816V KIT mutation with allele-specific ARMS–RT-qPCR, and multiparameter flow cytometric evaluation of BM MC, as previously reported [5,19,20].

For the purpose of the present analysis, we focused on all the consecutive adult patients with a confirmed diagnosis of mastocytosis or other CMD residents in the Veneto region, referred to our institution from January 2006 to January 2021. The diagnosis of CM or SM variants was established according to the WHO criteria [1,2]. ISM and SSM were considered as non-advanced variants, while ASM, SM-AHN, and MCL were considered as advanced variants of SM. Patients with maculo-papular CM (MPCM) who did not perform BM evaluation were defined as having mastocytosis in the skin (MIS) [1]. Symptomatic patients without skin lesions who did not meet sufficient WHO criteria for SM diagnosis, but in whom the presence of D816V *KIT* mutation and/or of BM MC with aberrant expression of CD25 and/or CD2 were demonstrated, were classified as having MMAS [3,4,5,6].

The prevalence of mastocytosis was determined as the proportion between patients affected by the disease resident at 1 January 2021 in Veneto and in Verona, respectively, and the total adult inhabitants reported at the same time in the same areas (4,231,643 and 797,459, respectively). The incidence of new cases/year was calculated on the data of the resident population from 2018 to 2020. Data about resident population were taken from the website https://dati.istat.it.

The presence of osteoporosis was evaluated with bone densitometry (DXA, Hologic QDR Delphi) at the lumbar spine (L1–L4) and at the total proximal hip. Radiological evaluation of total spine (X-ray) was performed for ascertaining the presence of vertebral fractures. Osteoporosis was defined according to the traditional WHO criteria, as a bone mineral density (BMD) T-score of −2.5 SD or less at lumbar spine or hip or by the presence of fragility fractures [16,21]. Since mastocytosis patients include pre-menopausal females and males < 50 years, we used also a Z-score threshold of −2.0 as a diagnostic criterion in order to defined low BMD.

Comparisons of factors were performed using χ^2^ test of categorical variables and Student *t*-test for continuous variables. Survival was estimated using the Kaplan–Meier estimator with overall survival (OS) being defined as the time between diagnosis and death (from any cause) or end of follow-up (censored) and progression free survival (PFS) as the time between diagnosis and shift to a higher-grade WHO variant of mastocytosis (i.e., from CM to SM, or from ISM to SSM or advanced forms) [13].

The study design adhered to the ethics principles of the Declaration of Helsinki and was approved by the ethics committee of the Azienda Ospedaliera Universitaria Integrata di Verona, Italy (protocol n° 1828 approved on 5 December 2010). Written informed consent was obtained from all the enrolled patients.

## 3. Results

From 1 January 2006 to 31 December 2020, 675 adult patients resident in Veneto and with suspicion of mastocytosis or with a diagnosis of mastocytosis already established were sent to the GISM for evaluation. The diagnosis of mastocytosis or other CMD was confirmed in 502 cases. The median age at diagnosis of the overall population with CMD was 54 years (range 16–86). There was a male predominance, with an M/F ratio of 1.5 (300/202).

### 3.1. Reason for Referral

The reasons for referral to GISM of these 502 patients were: severe HVA (*n* = 243), reactions after non-Hymenoptera insect sting (*n* = 6), drugs or food allergy (*n* = 17), recurrent MC-mediator symptoms (*n* = 6), diagnosis of MPCM (*n* = 162), unexplained osteoporosis (*n* = 35), osteosclerosis (*n* = 3), and hematological disorders (*n* = 30).

### 3.2. Final Diagnosis

The final diagnosis was CM in 19 cases (4%), ISM in 392 (78%), SSM in 6 (1%), SM-AHN in 27 (5%), and ASM in 6 (1%). Within the group of ISM patients, 156 had typical skin lesions (ISMs+) (40%), while the other 236 (60%) were diagnosed as BMM. In this series there were no cases of MCL or MCS. Among patients with SM-AHN the majority of associated neoplasia were of myeloid origin (81.5% of cases), namely, myeloproliferative neoplasia (*n* = 8), chronic myelomonocytic leukemia (*n* = 9), myelodysplastic/myeloproliferative neoplasia (*n* = 3), myelodysplasia (*n* = 2). Lymphoid neoplasia was diagnosed in 18.5% of cases, namely, multiple myeloma (*n* = 1), chronic lymphocytic leukemia (*n* = 1), and non-Hodgkin Lymphomas (*n* = 3).

In 23 patients (5%) with MPCM, BM evaluation was not performed either due to patient refusal or clinical decision (very low tryptase, no symptoms) and they were classified as MIS [1]. In 29 patients (6%) without skin lesions the major criterion was not demonstrated, and less than 3 minor WHO criteria were met, even though they had the D816V KIT mutation and/or the presence of MC with aberrant expression of CD25/CD2 in the BM: in these cases, the final diagnosis was MMAS.

### 3.3. Prevalence and Incidence of CMD

The prevalence of all types of CMD in the adult population resident in the Veneto region and in the Verona province at January 2021 was 11.2 and 20.6 per 100,000 inhabitants, respectively. The prevalence of SM in the adult population of the Veneto region and in the Verona province at the same time was 10.2 and 17.2 per 100,000 inhabitants, respectively.

The number of new diagnoses of all types of CMD from January 2006 to January 2021 in the people resident in the province of Verona and in the other provinces of the Veneto region (excluding Verona) is represented in Figure 1. In the Verona province, the number of new CMD diagnoses/year appears stable since 2012 with a mean number of 11.8 cases per year (±3.6 SD), while in the other provinces, the number of new cases/year referred to GISM progressively increased from 5 to 32 new diagnosis/year (2006–2021). The incidence of new cases/100,000 inhabitants/year in the adult population residing in the province of Verona in 2018, 2019 and 2020 was 1.26, 0.75, and 1.25, respectively (mean 1.09).

In Table 1, we compared the frequency of the SM variants diagnosed at GISM with the main series published in the literature. Within our series, the comparison of the frequency of variants between the patients from the province of Verona alone and the entire Veneto region did not show any significant difference. In the Veneto cohort, ISM represented the great majority (91%) of the SM, and BMM cases accounted for 54.8% of all the SM variants. Among the advanced forms, the most frequent was SM-AHN (6.3%). As shown in Figure 2, there was a steep increase in the number of BMM diagnoses comparing the first years of GISM activity (before 2011) and the more recent years. Similarly, only 2 diagnoses of SM-AHN were made before 2011, while 8 and 17 cases were recognized in the years 2011–2015 and 2016–2020, respectively.

Familial mastocytosis was reported in 16 patients (3.0%). Fifty patients (10%) reported family history of other hematological disorders, namely, leukemias (*n* = 18), lymphomas (*n* = 13), multiple myeloma (*n* = 5), unspecified anemias (*n* = 5), and myelodysplastic syndromes (*n* = 4).

### 3.4. Clinical, Laboratory, and Diagnostic Characteristics of Patients

The main clinical, laboratory, and diagnostic data of patients according to the different CMD are detailed in Table 2.

The median age was significantly lower in non-advanced SM variants (53 years, range 18–86) than in ASM and SM-AHN (69 years, range 35–80) (*p* < 0.001).

The M/F ratio was lower in non-advanced mastocytosis variants with skin involvement (CM, MIS, ISMs+, and SSM: 0.74, 0.77, 0.9, and 1.0, respectively) than in BMM (2.0) and in advanced SM (ASM and SM-AHN: 2.0 and 1.5, respectively). MMAS patients showed the highest male prevalence (4.8).

Skin lesions were present in 216 (43%) of the total 502 cases: considering gender distribution irrespective of final diagnosis, skin involvement was significantly more frequent in the female population (55%, 111/202) than in males (35%, 105/300) (*p* < 0.001).

Diagnosis of SM was less frequently based on the detection of the major WHO criterion in non-advanced (45%, 179/398) than in advanced SM (73%, 24/33) (*p* = 0.002). Of note, the diagnosis of BMM was based only on minor criteria in 64% of cases.

Serum tryptase was >20 ng/mL in 217 out of 493 patients with available data (44%), and specifically in all cases of ASM and SSM, in 63% of patients with SM-AHN, in 58% of patients with ISMs+, in 39% of patients with BMM, in 10% of patients with MMAS, and in 4.3% of patients with MIS. A tryptase value higher than or equal to 200 ng/mL was found in 3% (14/493) of cases, and particularly in 5 out of 6 cases of SSM. Serum tryptase was <20 ng/mL in all cases of CM.

The KIT D816V mutation was found in 411 out of 454 (90.5%) SM patients evaluated on BM and/or peripheral blood. In two cases other mutations of exon 816 of the KIT gene were detected (one D816H and one D816G). D816V KIT mutation was less frequently detected in advanced SM compared to non-advanced variants (*p* < 0.001) Moreover, also the aberrant expression of CD25 and/or CD2 was more frequent in non-advanced SM than in advanced ones (*p* < 0.001). Atypical morphology was detected in >25% of MC on BM smears in 354 out of 394 evaluable SM patients.

Karyotype was performed in 58 patients (2 CM, 2 MMAS, 23 BMM, 23 ISMs+, 2 SSM and 6 SM-AHN). Abnormal karyotype was documented only in two patients with SM-AHN (chromosome 7 monosomy and chromosome 8 trisomy, respectively) and in three patients with BMM (all were old males with age-related loss of chromosome Y).

NGS study for other myeloid genes was performed in 17 patients (3 SSM, 3 ASM, and 11 SM-AHN): all but one had at least one additional mutation, of which the more frequent were *JAK2 V617F* (*n* = 5), *TET2* (*n* = 4), *SF3B1* (*n* = 2), *NRAS* (*n* = 2), and *ASXL1* (*n* = 2) mutations.

Organomegaly was documented in 13% of patients (*n* = 64), namely, splenomegaly 7% (*n* = 33), hepatomegaly 8% (*n* = 38), and lymphadenomegaly 3% (*n* = 15). Seventeen patients (3.4%) had at least two of these three clinical features. These findings were documented more frequently in advanced SM variants (*p* < 0.001).

### 3.5. Hymenoptera Venom Allergy and Other Allergic Reactions

In the whole population, an history of allergic reactions was reported in 316 patients (63%). Detailed triggers of allergic reactions according to the CMD type are reported in Table 3. Severe HVA was reported in 251/502 patients (50%). Specifically, HVA was less frequently reported in patients with CM (5%), MIS (9%), SM-AHN (15%), ASM (17%), and ISMs+ (22%). Conversely, it represented the main symptom that led to diagnosis in 78% of patients with BMM and in 87% of patients with MMAS. (Table 3) In the majority of patients with severe HVA (232/251; 92.4%) VIT could be started after specific serologic and skin tests, or continued if already ongoing with the indication to pursue it long life. VIT was well tolerated in all cases. In about 8% of patients, VIT could be not administrated due to negative or not conclusive tests. Twenty-two out of 232 (9.5%) patients undergoing VIT developed further allergic reactions after recurrent Hymenoptera sting: in these cases, VIT directed against another insect was added or an increased dosage of the same venom was administered, according to the results of the specific tests. No fatal events were reported.

Allergic reactions after ingestion of drugs or food were reported in 53 (10.6%) and 24 cases (4.8%), respectively; in 7 cases (1.4%) severe reaction after sting of Dyptera insects was reported. In 18 cases more than 1 type of trigger was reported.

According to the international guidelines [25], all patients with diagnosis of SM received two packs of self-injectable adrenaline and instruction on their use; counseling on general behaviors and specific information on the use of anesthetic drugs were given. Allergic tests for drugs or food were performed when indicated.

### 3.6. Bone Involvement

Data on bone involvement at diagnosis were available in 459 patients; 275 males and 184 females. Osteopenia was documented in 147 (32.0%) and osteoporosis in 161 cases (35.1%), respectively. Fragility fractures already documented or asymptomatic and/or paucisymptomatic fractures identified at the first observation at GISM were recorded in 107 out 161 (66.5%) patients with osteoporosis. (Table 4). The majority of the fractures involved the spine (92%), and in 69% of cases at least two vertebral bodies were involved (up to 9). Rib and femoral fractures were documented in 7% and 2% of patients with osteoporosis, respectively.

The frequency of osteoporosis in the whole population was higher in females than in males, (40% and 32% respectively), even though the difference was of marginal statistical significance (*p* = 0.059) (Table 4). Notably, osteoporosis had not previously been diagnosed in 43.3% of females and in 81.1% of males. Moreover, in people aged 50 years or less osteoporosis was documented more frequently in males (24%) than in females (16%), even though without reaching significant difference (*p* = 0.182). The prevalence of fragility fractures was significantly higher in young males (12%) than in young females (2.5%) (*p* = 0.013) (Table 4).

Data about bone involvement according to CMD were reported in the Table 5. The frequency of osteoporosis at diagnosis according to the CMD type was as follows: BMM 93/223 (41.7%), ISMs+ 43/149 (28.8%), SM-AHN 11/20 (55.0%), ASM 2/5 (40.0%), CM 4/19 (21.1%), MIS 5/13 (38.5%), and MMAS 4/26 (15.4%). Diffuse osteosclerosis was documented in 15 cases (3.2%), namely, 2 patients with BMM, 4 ISMs+, 1 ASM, 2 SM-AHN, and all 6 patients with SSM. In 5 of the 11 cases osteosclerosis was associated with diffuse minute osteolysis. In other 30 patients (6.5%) focal areas of osteosclerosis were documented.

After rheumatological evaluation of these 459 patients, the following therapy was prescribed: vitamin D supplementation (69% of cases), calcium supplementation (4%), parenteral infusions of zoledronate every 12–18 months (24%), denosumab (2%), and oral bisphosphonates (4%). In half of these latter cases, treatment was subsequently replaced by zoledronate or denosumab. Four patients complaining of important painful symptoms and with high number of fractures at disease onset received pegylate alpha-interferon 90 mcg/week for one year associated with zoledronate.

### 3.7. Treatment of Advanced SM

Eight out of 33 patients with advanced SM were treated with MC-directed cytoreductive therapy, as follows: imatinib (*n* = 1), midostaurin (*n* = 3), avapritinib (*n* = 2), cladribine (*n* = 1), alpha-interferon (*n* = 2). One patient with ASM received four lines of treatment, including cladribine, midostaurin, ripretinib, and presently avapritinib.

Fourteen patients with SM-AHN received cytoreductive treatment for their AHN, namely, hydroxyurea (*n* = 7), azacytidine (*n* = 2), ruxolitinib (*n* = 1), chemotherapy for lymphoma (*n* = 3) or multiple myeloma (*n* = 1).

The remaining ten patients did not need cytoreduction, including a patient with ASM who refused any treatment.

### 3.8. Survival

In the overall cohort, the median follow-up from the first observation by the GISM was 62 months (range 0–187) and the median follow-up from diagnosis was 72 months (range 0–613).

Twenty-six patients (5.2% of the whole cohort and 6% of the patients with SM) died during follow-up. No deaths were recorded in patients with CM, MIS, or MMAS. Causes of death were: disease progression (*n* = 6), another malignancy (*n* = 5), pneumonia (*n* = 1), cardiovascular causes (*n* = 4), liver cirrhosis (*n* = 1), and suicide (*n* = 1). In eight cases the cause of death was unknown.

Limiting the analysis to patients with SM, the estimated 10-year OS was more favorable in non-advanced variants (BMM, ISMs+, and SSM: 92.3%, 94.9%, and 100%, respectively) than in advanced forms (ASM and SM-AHN: 83.3% and 60.9%, respectively) (Figure 3). There were no differences in OS according to gender in the overall population and in the different variants (data not shown).

During follow-up, 3 of 19 patients with CM were diagnosed with ISMs+ after 122, 86, and 99 months, respectively. In the latter case, there was a further progression to SSM after 30 months. Other 14 patients (1 MIS, 2 MMAS, 1 BMM, 3 ISMs+, 2 SSM, 2 ASM, and 3 SM-AHN) progressed to advanced forms of SM, namely, 1 to ASM, 2 to MCL, and 11 to SM-AHN.

The PFS of the individual SM variants is represented in Figure 3. The estimated 10-year PFS was 100% in BMM, 98.1% in ISMs+, 62.5% in SSM, 55.6% in ASM, and 87.4% in SM-AHN.

## 4. Discussion

The prevalence of mastocytosis is believed to have increased significantly in the past 15–20 years after the definition of the WHO diagnostic criteria, the development of more sensitive methods for detecting MC clonality (e.g., molecular methods for D816V *KIT* mutation, multiparametric flow cytometric analysis), the growing experience of physicians and pathologists and the creation of multidisciplinary centers specialized in this disease.

A study conducted on the adult population of the Groningen region including 48 patients with ISM, SSM, and Urticaria Pigmentosa (UP) found a prevalence of about 13 per 100,000 inhabitants [12]. In 2014, another study based on the Danish health registry and conducted on a national cohort including 548 patients with any type of SM and UP with a minimum age of 15 years, estimated an average incidence of 0.89 per 100,000 people per year and a prevalence of 9.59 cases per 100,000 inhabitants [11]. More recent data about of epidemiology of mastocytosis are lacking. Moreover, the reported frequency of the different mastocytosis variants differs according to the type of center.

Here, we reported the data of prevalence, incidence, and frequency of variants in the Verona province and Veneto region based on the activity of a multidisciplinary center that applies the best diagnostic methods recommended by the international guidelines.

Based on the activity of our reference center, in the adult population of the Veneto region, we were able to demonstrate a prevalence of 12 per 100,000 inhabitants for all types of mastocytosis and other CMD, and of 10.2 per 100,000 inhabitants when considering SM alone, in accordance with the already published data [11]. In addition, when the observation was restricted to the province of Verona, the prevalence rate increased to 20.6 and 17.2 per 100,000 inhabitants for all types of mastocytosis and for SM alone, respectively. Additionally, the mean incidence of new cases in the province of Verona was slightly higher than previously reported (1.09 new cases/100,000 inhabitants/year). These results are correlated to a very close collaboration with all Allergological, Dermatological, Rheumatological, and Hematological Units in the province, which leads us to assume that virtually all the patients with suspected mastocytosis are referred to us. The frequency of ISM assessed in such a “real life” setting with a multidisciplinary approach appears to be the highest reported to date, close to 90%. Moreover, in more than half of the cases ISM had the characteristics of BMM, confirming the underestimation of this provisional WHO variant [7,8]. The number of new diagnoses of BMM in our series progressively increased in the last 15 years in accordance with the greater diffusion of knowledge among specialists about the association of BMM with heterogeneous clinical scenarios, particularly severe unexplained osteoporosis, and severe allergic reactions especially after Hymenoptera sting. The application of predictive CMD scores such as the REMA (Red Española de Mastocitosis) score has been very helpful in identifying patients suffering from mediator symptoms who need BM evaluation [26]. It should also be emphasized that in 64% of cases, the diagnosis of BMM was only performed on the basis of minor criteria, and therefore it was necessarily dependent on the use of sensitive molecular biology techniques for D816V *KIT* mutation detection associated with multiparametric flow cytometric study and to histological examination of the BM biopsy by an expert pathologist.

We could confirm in our series that advanced variants of SM are associated with older age and male sex, in line with previous reports [27,28], but their frequency was less than 9% of cases, much lower than reported in the past [10], and also lower than recently reported in the ECNM registry [13]. Of interest, the number of new diagnoses of SM-AHN progressively increased from 2006 to 2021, along with a greater experience of the dedicated pathologist, thus confirming that SM-AHN is still an underestimated variant, often diagnosed with a substantial delay [28].

Among non-advanced SM, younger age and female sex are associated with the presence of skin lesions, while BMM patients are predominantly males and older. The absence of skin lesions likely justify why BMM are diagnosed at a later time, while we could not find a firm explanation for the prevalence of male sex, but we hypothesized that males could be more exposed for professional or recreational reasons to Hymenoptera stings.

In our series HVA represented the main circumstance that led to the diagnosis of CMD. Moreover, HVA was mainly associated with ISM, above all BMM, and MMAS, as previously reported, and was rarely reported in CM and in advanced SM (SM-AHN and ASM). Early diagnosis of occult BMM in patients with HVA allowed for administration of long life VIT and helped to prevent further severe reactions, in some case lethal. VIT was well tolerated, only less than 10% of patients were not fully protected, similar to that reported in patients without SM; in these cases, additional VIT or increased dose of VIT was administrated on the basis of allergic tests [29,30,31].

Osteoporosis and fragility fractures represented a frequent and sometimes very disabling complication of CMD, with a frequency of 35.1% and 23.3%, respectively, in the whole cohort. In patients aged >50 years osteoporosis was more frequent in females, accounting a quote of post-menopausal osteoporosis. (*p* < 0.001). Notably the diagnosis of mastocytosis allowed ascertaining a previously unknown osteoporosis in about 81.1% of males and in 43.3% of females. Moreover, as previously reported, osteoporosis and fragility fractures were documented in a non-negligible proportion of young subjects, especially males [16]. In our series, in fact, osteoporosis was reported in 24% of young males and in 16% in females, and fragility fractures were significantly more frequent in males (12%), than in females (2.5%). Osteosclerosis, especially diffuse osteosclerosis, is a rarer event, as is the presence of small osteolysis, and these conditions are associated with SMM and advanced variants of SM.

The OS and PFS data of our cohort confirm that the non-advanced variants of SM have an overall good prognosis, in agreement with the data reported in the literature, while the prognosis of the few advanced forms was better than expected [10,13]. Progressions were exceedingly rare in ISM patients, and the difference in OS between patients with ISMs+ and BMM, which seemed to be more pronounced after 10 years of follow-up, albeit not statistically significant, was likely to be related to the median age at diagnosis, which was more than a decade lower in patients with ISMs+ as compared to BMM.

## 5. Conclusions

Our study confirms that a multidisciplinary approach and the creation of a network of local specialists are necessary to suspect and diagnose mastocytosis and other CMD in adults and to manage the complex problems of patients suffering from these diseases. This approach allowed an increase in the number of diagnoses, therefore demonstrating a greater prevalence and incidence of CMD in the province of Verona, especially of the indolent forms. In particular, we have demonstrated a much higher prevalence than that reported so far of the BMM variant, confirming that this form is underestimated. We have also shown how the availability of sensitive diagnostic techniques in the reference center together with the collaboration of expert pathologists is fundamental in the diagnosis of CMD. This condition was essential to diagnose most cases of BMM thanks to the minor criteria only, and to increase over time the number of new diagnoses of SM-AHN.

The correct and timely diagnosis of CMD, especially in the indolent forms, has to be considered of fundamental importance; in fact, it allows discovering and treating underdiagnosed osteoporosis, especially in males, a complication that is not infrequent even in young subjects, with sometimes disabling consequences. Moreover, diagnosis of CMD in patients with HVA allows the lifelong continuation of VIT in order to prevent further severe, even fatal, allergic reactions.

In conclusion, a multidisciplinary diagnostic approach and a multidisciplinary treatment strategy are of outstanding importance for adult patients with mastocytosis and require the presence of specialized territorial centers.

## Figures and Tables

**Figure 1 cancers-13-06380-f001:**
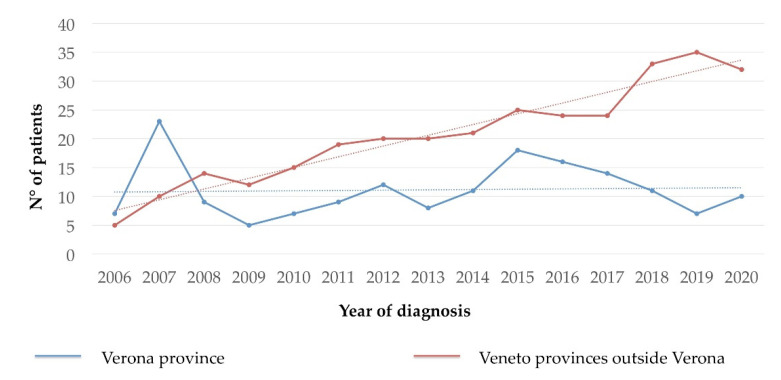
Number of new diagnoses of clonal mast cell disorders (all types) per year in the population residing in the province of Verona and in the other provinces of the Veneto region (excluding Verona).

**Figure 2 cancers-13-06380-f002:**
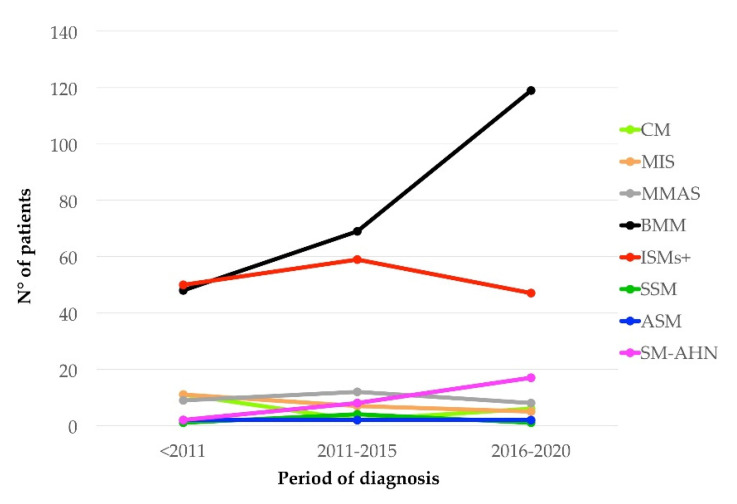
Number of new diagnoses of clonal mast cell disorders according to disease subtype and period of activity. Legend: CM: cutaneous mastocytosis; MIS: mastocytosis in the skin; MMAS: monoclonal MC activation syndromes; BMM: bone marrow mastocytosis; ISMs+, indolent systemic mastocytosis with skin lesions; SSM: smoldering systemic mastocytosis; ASM, aggressive systemic mastocytosis; SM-AHN, systemic mastocytosis with associated hematological neoplasm.

**Figure 3 cancers-13-06380-f003:**
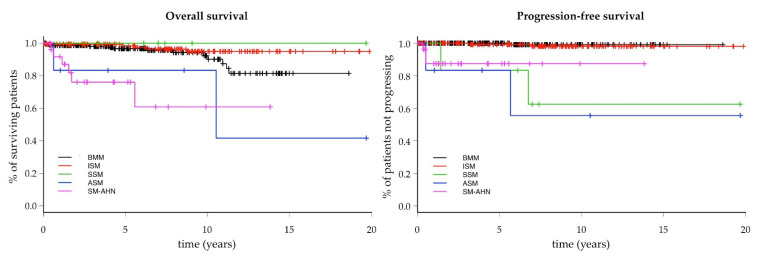
Overall (**left** panel) and Progression-Free (**right** panel) survival of the patients with systemic mastocytosis, according to the WHO variant. Legend: BMM: bone marrow mastocytosis; ISMs+, indolent systemic mastocytosis with skin lesions; SSM: smoldering systemic mastocytosis; ASM, aggressive systemic mastocytosis; SM-AHN, systemic mastocytosis with associated hematological neoplasm.

**Table 1 cancers-13-06380-t001:** Comparison between the frequencies of the variants of systemic mastocytosis in the present study and those reported in the main series of literature.

Systemic Mastocytosis	Lim et al. [10] *n* (%)	Sanchez et al. [22] *n* (%)	Wimazal et al. [23] *n* (%)	Cohen et al. [11] *n* (%)	Pieri et al. [24] *n* (%)	Sperr et al. [13] *n* (%)	GISM Veneto (2021)	GISM Verona (2021)	Veneto vs. Verona (*p*)
ISMs+ n°	159	93	81	450	418	1006	392	127	ns
%	(46.5)	(82.3)	(82.6)	(82.1)	(89.1)	(76.3)	(91.0)	(89.4)	
BMM n°	36	16	nr	nr	165	268	236	77	ns
%	(10.5)	(14.1)			(35.9)	(17.4)	(54.8)	(54.2)	
SSM n°	22	nr	7	nr	20	53	6	3	ns
%	(6.4)		(7.1)		(4.3)	(4.0)	(1.4)	(2.1)	
ASM n°	41	11	5	8	28	62	6	2	ns
%	(12.0)	(9.7)	(5.1)	(1.4)	(6.1)	(4.7)	(1.4)	(1.4)	
SM-AHN n°	138	6	11	24	21	174	27	10	ns
%	(40.4)	(5.3)	(11.2)	(4.4)	(4.6)	(13.2)	(6.3)	(7.0)	
MCL n°	4	2	1	5	1	23	0	0	ns
%	(1.2)	(1.8)	(1.0)	(0.9)	(0.2)	(1.7)	(0)	(0)	
total	342	113	98	548	460	1318	431	142	

Legend: CM: cutaneous mastocytosis; MIS: mastocytosis in the skin; MMAS: monoclonal MC activation syndrome; BMM: bone marrow mastocytosis; ISMs+: indolent systemic mastocytosis with skin lesions; ASM: aggressive systemic mastocytosis; SM-AHN: systemic mastocytosis with associated hematological neoplasm; nr: not reported; ns: not significant.

**Table 2 cancers-13-06380-t002:** Characteristics of patients with clonal mast cell disorders.

Final Diagnosis	CM	MIS	MMAS	BMM	ISMs+	SSM	ASM	SM-AHN
Total cases	19	23	29	236	156	6	6	27
Median age, years (range)	43 (22–74)	29 (18–69)	54 (32–74)	57 (20–86)	44 (18–80)	64 (46–79)	65 (51–76)	69 (35–80)
Male/Female ratio	0.73	0.77	4.8	2.0	0.90	1.0	2.0	1.5
Skin involvement n° (%)	19 (100)	23 (100)	0 (0)	0 (0)	156 (100)	6 (100)	2 (33.3)	9 (33.3)
Serum Tryptase								
Median (ng/mL)	7.5	4.3	11.0	17.4	25.5	266.5	185.0	28.0
range	4.4–14.6	(2.4–68)	(3.4–27)	(3.6–134)	(2.9–582)	(168–377)	(102–589)	(7.9–761)
>20 ng/mL (%)	0 (0)	1 (4.3)	3 (10.3)	93 (39.4)	91 (58.3)	6 (100)	6 (100)	17 (63.0)
Major WHO SM criteria n° (%)	0 (0)	nd	0(0)	85 (36)	88 (56)	6 (100)	6 (100)	18 (67)
Exon 816 *KIT* Mutation, n° (%)	0 (0)	nd	15 (52)	224 (95)	148 (95)	6 (100)	4/5 (80)	18/23 (78)
CD25+ and/or CD2+ BM MC; n° (%)	0 (0)	nd	15 (52)	215 (91)	127 (81)	6 (100)	4/5 (80)	18/23 (67)
>25% BM atypical MCs; n° (%)	0 (0)	nd	9 (31)	204 (86)	122 (78)	3/6 (50)	5/5 (100)	17/25 (68)
Organomegaly n° (%)								
- Splenomegaly	1 (0)	0 (0)	1 (3)	5 (2)	6 (4)	2 (33)	5 (83)	13 (48)
- Hepatomegaly	0 (0)	0 (0)	0 (0)	14 (6)	12 (8)	0 (0)	3 (50)	9 (33)
- Lymphadenopathy	0 (0)	2 (9)	0 (0)	1(0,4)	2 (1)	1 (17)	2 (33)	7 (26)

Legend: BM: bone marrow, MC: mast cell; CM: cutaneous mastocytosis; MIS: mastocytosis in the skin; MMAS: monoclonal MC activation syndrome; BMM: bone marrow mastocytosis; ISMs+: indolent systemic mastocytosis with skin lesions; ASM: aggressive systemic mastocytosis; SM-AHN: systemic mastocytosis with associated hematological neoplasm; nd: not done

**Table 3 cancers-13-06380-t003:** Allergic reactions according to the diagnosis of clonal mast cell disorder.

Final Diagnosis	CM	MIS	MMAS	BMM	ISMs+	SSM	ASM	SM-AHN
HVA, n° (%)	1 (5)	2 (9)	25 (86)	184 (78)	34 (22)	0 (0)	1 (17)	4 (15)
Other insects n° (%)	0 (0)	0 (0)	1 (3.4)	3 (1.3)	2 (1.3)	1 (17)	0 (0)	0 (0)
Drug, n° (%)	1 (5)	1 (4)	0 (0)	22 (9)	20 (13)	1 (17)	2 (33)	6 (22)
Food, n° (%)	0 (0)	3 (13)	1 (3)	14 (6)	6 (4)	0 (0)	0 (0)	0 (0)

Legend: HVA: Hymenoptera venom allergy; CM: cutaneous mastocytosis; MIS: mastocytosis in the skin; MMAS: monoclonal MC activation syndrome; BMM: bone marrow mastocytosis; ISMs+: indolent systemic mastocytosis with skin lesions; ASM: aggressive systemic mastocytosis; SM-AHN: systemic mastocytosis with associated hematological neoplasm.

**Table 4 cancers-13-06380-t004:** Bone involvement in clonal mast cell disorders according to gender and age.

Type of Bone Involvement	Total n° (%)	Median Age, y (Range)	Males >50 y n° (%)	Females >50 y n° (%)	*p*-Value (Males vs. Females) >50 y	Males ≤ 50 y n° (%)	Females ≤50 y n° (%)	*p*-Value (Males vs. Females) ≤50 y
Evaluated patients	459		162	103		113	81	
Osteopenia	147 (32)	52 (20–80)	50 (31)	30 (29)	0.763	39 (35)	28 (35)	0.993
Osteoporosis	161 (35)	58 (20–86)	60 (37)	61 (59)	<0.001	27 (24)	13 (16)	0.182
Fragility fractures	107 (23)	61 (39–86)	45 (28)	46 (45)	0.004	14 (12)	2 (2.5)	0.013
Osteosclerosis								
- diffuse	15 (3)	60 (35–79)	6 (3.7)	5 (4.9)	0.647	1 (0.8)	3 (3.7)	0.170
- patchy	26 (6)	54 (23–74)	15 (9.3)	3 (2.9)	0.453	3 (2.7)	5 (6.2)	0.224

**Table 5 cancers-13-06380-t005:** Bone involvement in patients according to the diagnosis of clonal mast cell disorder.

Final Diagnosis	CM	MIS	MMAS	BMM	ISMs+	SSM	ASM	SM-AHN
Evaluable patients	19	13	24	223	149	6	5	20
Osteopenia	5 (26)	3 (23)	6 (25)	71 (32)	57 (39)	0 (0)	1 (20)	4 (19)
Osteoporosis	4 (21)	4 (31)	4 (17)	93 (42)	43 (29)	0 (0)	2 (40)	11 (55)
Fragility fractures	1 (5)	1 (8)	3 (12.5)	73 (33)	18 (12)	1 (17)	2 (40)	9 (43)
Osteosclerosis	1 (5)	1 (8)	0 (0)	15 (7)	13 (9)	6 (100)	2 (40)	3 (14)
focal	1	1	0	13	9	0	1	1
diffuse	0	0	0	2	4	6	1	2
Small osteolysis	0 (0)	0 (0)	0 (0)	0 (0)	0 (0)	2 (33)	0 (0)	2 (10)
Large osteolysis	0 (0)	0 (0)	0 (0)	0 (0)	0 (0)	0 (0)	0 (0)	1 (5)

Legend: CM: cutaneous mastocytosis; MIS: mastocytosis in the skin; MMAS: monoclonal MC activation syndrome; BMM: bone marrow mastocytosis; ISMs+: indolent systemic mastocytosis with skin lesions; ASM: aggressive systemic mastocytosis; SM-AHN: systemic mastocytosis with associated hematological neoplasm.

## Data Availability

The data that support the findings of this study are available from the corresponding author upon reasonable request.

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
