# Peer review of "A Multidisciplinary Diagnostic Approach Reveals a Higher Prevalence of Indolent Systemic Mastocytosis: 15-Years’ Experience of the GISM Network"

_cancers, 2021, doi:10.3390/cancers13246380_

Round 1

Reviewer 1 Report

In this study, Zanotti et al determined the prevalence and incidence of clonal mast cell diseases in Venoto region and Verona. They also describe the clinical characteristics of their 502 patients.

This work is very interesting because it precise the prevalence of such a rare disease. It also demonstrates that an interdisciplinarity allows to better identify patients with systemic mastocytosis (in particular ISM patients). Such an organization (along with an improvement in laboratory testing) lead to an improved detection of patients with bone marrow mastocytosis.

The paper is well written and organized. I just would like to have more details about the genetic characteristics of the patients (presence of additional mutation if karyotype and NGS sequencing have been performed). I would also appreciate if the author could give more details about the treatments used in their patients to target the clonal mast cells. These informations from a "real life" cohort could be valuable to determine the impact on survival and the evolution of the disease.

Author Response

In this study, Zanotti et al determined the prevalence and incidence of clonal mast cell diseases in Veneto region and Verona. They also describe the clinical characteristics of their 502 patients.

This work is very interesting because it precise the prevalence of such a rare disease. It also demonstrates that an interdisciplinarity allows to better identify patients with systemic mastocytosis (in particular ISM patients). Such an organization (along with an improvement in laboratory testing) lead to an improved detection of patients with bone marrow mastocytosis.

The paper is well written and organized.

We thank this reviewer for the positive comments and we appreciate that the main message of our work has been fully captured.

Comment: I just would like to have more details about the genetic characteristics of the patients (presence of additional mutation if karyotype and NGS sequencing have been performed). I would also appreciate if the author could give more details about the treatments used in their patients to target the clonal mast cells. These informations from a "real life" cohort could be valuable to determine the impact on survival and the evolution of the disease.

Response:Details about genetic and molecular features have been added to the section 3.4. A novel section (i.e. 3.7) has been added to describe in details the treatment of advanced forms.  

Reviewer 2 Report

The manuscript presents a well described mastocytosis cohort. Despite large, multi-center cohorts beeing published within the ECNM, this study adds significantly to the field. Overall, the paper is well written and results are clearly presented. I only have two points to consider:

1) I got puzzled with the survival data presented in Figure 3. While Progression-free survival (PFS) is typically defined as the time to progression or death from any cause, this study uses a different definition only accounting for progression but not for death. In my mind, another less misleading term should be used for the data presented in the right panel. Alternatively, PFS accoriding to standard definitions could be shown.

2) What is the explanation of the differences in overall survial between BMM and ISM (Figure 3, left panel)? Is there a difference in age distribution between these two subtypes that is not reflected by the median age?

Author Response

We thank this reviewer for providing helpful comments. This allows us to better clarify the terminology and the meaning of our data

Comment: I got puzzled with the survival data presented in Figure 3. While Progression-free survival (PFS) is typically defined as the time to progression or death from any cause, this study uses a different definition only accounting for progression but not for death. In my mind, another less misleading term should be used for the data presented in the right panel. Alternatively, PFS accoriding to standard definitions could be shown.

Response: We adopted a stringent definition of PFS accounting as events only the shifts from a lower to a higher risk category of mastocytosis. The same definition was adopted by the ECNM (e.g. Sperr et al. Lancet Hematol 2019, a paper specifically dealing with prognostication), and we added this reference in the Method section. We opted to present the data in such a manner instead of a composite endpoint of progressions and deaths (i.e. event-free survival) to highlight that ISM and BMM have a very low risk of evolution, and deaths are related to other causes. This information is of value in the counseling of patients with indolent SM.

Comment: What is the explanation of the differences in overall survial between BMM and ISM (Figure 3, left panel)? Is there a difference in age distribution between these two subtypes that is not reflected by the median age?

Response: We thank the referee for capturing this inconsistence in our data. There was a typo in the median age of ISMs+ patients, which is 44 instead of 54, actually. This difference of more than a decade in age at diagnosis may explain the shape of survival curves between ISMs+ and BMM, even though the difference is not statistically significant. We apologize for the mistake and we seized the opportunity to address this question in the discussion.

Reviewer 3 Report

Zanotti et al. analyzed the diagnoses and findings of 502 adult patients with mastocytosis or other clonal mast cell disorders referred to the italian center of the Interdisciplinary Group for Study of Mastocytosis (GISM) in Verona from 2006 to 2020. Using a multidisciplinary diagnosis approach, they have provided an interesting and relevant study. In addition to the greater prevalence and incidence of systemic mastocytosis, they found that systemic mastocytosis was associated with hymenoptera venom allergy (BMM and MMAS) and osteoporosis even in young and male patients. The article reads well, however, I have a few specific comments that I would like to see addressed.

Major comments:

  1. Table 5, there are discrepancies between the osteroporosis numbers in the table and the ones in the text (eg. for BMM, ISMs+ ...). Which ones are correct? Please clarify.
  2. Authors showed that younger age and female sex are associated with the presence of skin lesions, while BMM patients are predominantly males and older. Is there any impact of sex on the overall survival in non-advanced variants (BMM, ISMs+, and SSM) and in advanced forms (ASM and SM-AHN)?

Minor points:

  1. Line 30 and 257: “clonal mast cells disorders” take out the “s”.
  2. Line 89: “fragility fractures. [15,16] and on administration of life-long venom immunotherapy…”, something wrong with the punctuation.
  3. Figure 1: indicate the legend on the X-axis and Y-axis.
  4. Table 1: column 2 and 8 titles not aligned to the others.
  5. Table 2: last row, numbers should be aligned in all columns.
  6. Figure 2: indicate the legend on the
  7. Line 245: “HVA was not reported in any patient with CM or ASM…” in the table 3, it says 1 for CM and ASM, please clarify.
  8. Line 308: “5.2% of the whole cohort an 6%…” something is wrong with this phrase, please rephrase.
  9. Table 4: in the column 2 title “totale” should be without “e”
  10. Table 5: numbers in column 2, 3 and 5 not perfectly aligned to the others.
  11. Figure 3: font size axis needs increases regarding labels within the figure.

Author Response

We thank this reviewer for the careful evaluation of our work. According to these comments and suggestions, we have revised our manuscript.

Comment: Table 5, there are discrepancies between the osteroporosis numbers in the table and the ones in the text (eg. for BMM, ISMs+ ...). Which ones are correct? Please clarify.

Response: These mistakes were amended and numbers are consistent now.

Comment: Authors showed that younger age and female sex are associated with the presence of skin lesions, while BMM patients are predominantly males and older. Is there any impact of sex on the overall survival in non-advanced variants (BMM, ISMs+, and SSM) and in advanced forms (ASM and SM-AHN)?

Response:There was no gender-related difference in OS both in the whole population and according to variants. We added this information in the results.

Minor points:

  1. Line 30 and 257: “clonal mast cells disorders” take out the “s”.
  2. Line 89: “fragility fractures. [15,16] and on administration of life-long venom immunotherapy…”, something wrong with the punctuation.
  3. Figure 1: indicate the legend on the X-axis and Y-axis.
  4. Table 1: column 2 and 8 titles not aligned to the others.
  5. Table 2: last row, numbers should be aligned in all columns.
  6. Figure 2: indicate the legend on the
  7. Line 245: “HVA was not reported in any patient with CM or ASM…” in the table 3, it says 1 for CM and ASM, please clarify.
  8. Line 308: “5.2% of the whole cohort an 6%…” something is wrong with this phrase, please rephrase.
  9. Table 4: in the column 2 title “totale” should be without “e”
  10. Table 5: numbers in column 2, 3 and 5 not perfectly aligned to the others.
  11. Figure 3: font size axis needs increases regarding labels within the figure.

Response: We corrected typos and improved the readability of figures as required

Round 2

Reviewer 3 Report

The authors improved the manuscript by answering my questions and requests.

However, numbers for MIS, MMAS and osteosclerosis in the text from line 305 to 308 were correct, no changes needed.

I would like to point that there are still some minor typos to be corrected: eg. in the minor point 1, Line 30 and 257: “clonal mast cells disorders” take out the “s”. The "s" was related to "cells" not to disorders.

Author Response

We thank this referee for the accurate revision. We amended the highlighted inconsistencies and typos.